# NMDA Receptor Glycine Binding Site Modulators for Prevention and Treatment of Ketamine Use Disorder

**DOI:** 10.3390/ph16060812

**Published:** 2023-05-30

**Authors:** Yu-Chin Hsiao, Mei-Yi Lee, Ming-Huan Chan, Hwei-Hsien Chen

**Affiliations:** 1Center for Neuropsychiatric Research, National Health Research Institutes, 35 Keyan Road, Zhunan, Miaoli County 35053, Taiwan; ttcc313@gmail.com (Y.-C.H.); meiyi070824@nhri.edu.tw (M.-Y.L.); 2Institute of Neuroscience, National Chengchi University, 64, Sec. 2, ZhiNan Road, Wenshan District, Taipei City 11605, Taiwan; minghuan@nccu.edu.tw; 3Research Center for Mind, Brain, and Learning, National Chengchi University, 64, Sec. 2, ZhiNan Road, Wenshan District, Taipei City 11605, Taiwan

**Keywords:** D-serine, sarcosine, breakpoint, reinstatement, sucrose

## Abstract

Ketamine offers a fast-acting approach to relieving treatment-resistant depression, but its abuse potential is an issue of concern. As ketamine is a noncompetitive N-methyl-D-aspartate receptor (NMDAR) ion channel blocker, modulation of NMDAR might be an effective strategy to counteract the abuse liability of ketamine and even to treat ketamine use disorder. This study evaluated whether NMDAR modulators that act on glycine binding sites can decrease motivation to obtain ketamine and reduce reinstatement to ketamine-seeking behavior. Two NMDAR modulators, D-serine and sarcosine were examined. Male Sprague–Dawley rats underwent training to acquire the ability to self-administer ketamine. The motivation to self-administer ketamine or sucrose pellets was examined under a progressive ratio (PR) schedule. The reinstatement of ketamine-seeking and sucrose pellet-seeking behaviors were assessed after extinction. The results showed that both D-serine and sarcosine significantly decreased the breakpoints for ketamine and prevented reinstatement of ketamine seeking. However, these modulators did not alter motivated behavior for sucrose pellets, the ability of the cue and sucrose pellets to reinstate sucrose-seeking behavior or spontaneous locomotor activity. These findings indicate that two NMDAR modulators can specifically reduce the measures of motivation and relapse for ketamine in rats, suggesting that targeting the glycine binding site of the NMDAR is a promising approach for preventing and treating ketamine use disorder.

## 1. Introduction

Ketamine is a frequently used anesthetic agent for both inducing and maintaining anesthesia. It also applies to pain management and treatment of depression. In addition, ketamine is effective in treating chronic posttraumatic stress disorder (PTSD) [1] and refractory anxiety [2,3]. Interestingly, ketamine shows great promise as an approach to treating addiction to certain drugs of abuse, such as cocaine or alcohol [4]. However, ketamine itself is a reinforcing stimulus, which can induce self-administration [5,6,7] and conditioned place preference [8,9] in animals. In fact, ketamine is classified as a controlled substance in many countries, owing to its putative addiction liability. The popularity of ketamine for recreational use has been increasing worldwide due to its reinforcing, hallucinogenic, tranquilizing, and dissociative effects. Ketamine use disorder has become a global public health issue [10].

The effects of ketamine in psychiatric disorders, such as depression, anxiety, and PTSD, generally last from a few days to 2 weeks, so repeated dosing is usually necessary to extend recovery. Although the doses of ketamine used for these diseases are extremely low compared with recreational use, the abuse risk still remains a concern. Therefore, it is crucial to explore complementary therapies that can enhance the therapeutic effectiveness of ketamine while minimizing its potential for abuse.

Blockade of NMDAR has been shown to link to the reinforcing effect of ketamine because noncompetitive NMDAR blockers, such as phencyclidine (PCP) and MK-801, also possess reward-related properties [11]. Modulating NMDAR, therefore, might be a potential strategy to counteract the abuse liability of ketamine. The present study assessed the ability of two NMDAR modulators, D-serine and sarcosine, to attenuate the reinforcing efficacy of ketamine.

D-serine is a co-agonist of the synaptic NMDAR glycine binding site. It has been demonstrated to alleviate certain behavioral effects of NMDAR blockers. Specifically, D-serine has been shown to decrease PCP and/or MK-801-induced hyperactivity, stereotyped behavior, ataxia [12], prepulse inhibition deficits [13], and cognitive deficits [14]. Although the interaction between D-serine and ketamine is limited, recent research indicates that D-serine can reverse the anti-tumor effect of ketamine [15]. Moreover, based on previous studies demonstrating a decline in D-serine levels in the nucleus accumbens of rats exposed to cocaine [16] or morphine [17], a potential link between D-serine and addiction has been proposed. Additionally, D-serine has been shown to reduce aversion-resistant alcohol drinking [18]. Collectively, D-serine may counteract not only the motivation for ketamine but also other addiction-like behaviors such as craving and relapse.

Sarcosine is a glycine transporter inhibitor and an NMDAR co-agonist [19]. Sarcosine has been shown to counteract the behavioral responses of noncompetitive NMDAR antagonists. The ketamine-induced PPI deficit and hyperlocomotion [20] and MK-801-induced impairment in the holeboard task, PPI task and the cue test of the fear conditioning task [21] were reversed by sarcosine. Moreover, sarcosine normalized MK-801-induced abnormal brain activity [21].

It is notable that, in addition to acting together with glutamate as mandatory co-agonists to regulate NMDAR activity, D-serine and sarcosine also regulate the trafficking of NMDAR [21,22]. Additionally, high concentrations of D-serine prime the NMDAR for endocytosis [23]. These modulatory effects of D-serine and sarcosine on NMDAR may be associated with their counteracting effects on ketamine abuse.

In this study, we utilized a self-administration model of ketamine addiction under a progressive ratio (PR) schedule, wherein the number of lever presses required for each successive reinforcer increased until responding ceased. The breakpoint, which is the number of responses required to obtain the final reward, is a well-established measure of the motivational strength and reinforcing efficacy of drugs. We determined whether D-serine and sarcosine could reduce the ketamine breakpoint first. Furthermore, we also assessed the potential of D-serine and sarcosine to prevent ketamine relapse by examining their effects on cue-induced and ketamine-induced reinstatement of ketamine-seeking behavior. Finally, we evaluated the selectivity of D-serine and sarcosine towards ketamine reinforcement by monitoring their effects on sucrose reinforcement and the reinstatement of palatable sucrose-seeking behaviors.

## 2. Results

### 2.1. Effects of D-Serine on Ketamine Self-Administration Using a Progressive Ratio (PR) Schedule

Initially, the effect of D-serine at 30 mg/kg on ketamine self-administration under a PR schedule was examined. No significant difference was observed between the D-serine (30 mg/kg) and the control group (Figure 1A). Subsequently, a higher dose of D-serine (100 mg/kg) was employed. As shown in Figure 1B, administration of D-serine (100 mg/kg) led to a significant decrease in the number of breakpoints (*t* = 2.947, *p* < 0.05), lever responses (*t* = 3.515, *p* < 0.05), and ketamine infusions (*t* = 3.114, *p* < 0.01).

### 2.2. Effects of Sarcosine on Ketamine Self-Administration Using a Progressive Ratio (PR) Schedule

A Latin square design was employed to determine the impact of two different doses of sarcosine (30 and 100 mg/kg) on ketamine self-administration under a PR schedule. One-way repeated measures ANOVAs were used to analyze the data (breakpoints: F_2,16_ = 14.004, *p* < 0.001; the number of lever responses: F_2,16_ = 12.975, *p* < 0.001; and ketamine infusions F_2,16_ = 16.536, *p* < 0.001). *Post hoc* comparisons showed that sarcosine dose dependently reduced the number of breakpoints, lever responses, and ketamine infusions (Figure 1C).

### 2.3. Effects of D-Serine on Reinstatement of Ketamine Seeking in Response to Cues and Drug Exposure

Following a stable response to ketamine self-administration, the rats underwent extinction training sessions that continued until the criteria were met. Due to individual variations, the number of days needed for each animal to achieve the extinction criterion differed. Nevertheless, no notable difference was observed in the number of lever responses during the final extinction session before each reinstatement.

D-serine (100 mg/kg) was administered 120 min prior to each reinstatement test. The data were analyzed using a two-way repeated measures ANOVA, revealing significant main effects for Cue (F_1,5_ = 63.208, *p* < 0.001) and D-serine treatment (F_1,5_ = 18.167, *p* < 0.01), as well as a significant interaction between cue and D-serine treatment (F_1,5_ = 20.557, *p* < 0.01). For ketamine priming-induced reinstatement ANOVA showed significant main effects for ketamine priming (F_1,5_ = 15.576, *p* = 0.011), but not for D-serine treatment (F_1,5_ = 3.007, *p* = 0.143); and no significant interaction between ketamine priming and D-serine treatment (F_1,5_ = 3.05, *p* = 0.141). *Post-hoc* tests indicated that D-serine (100 mg/kg) significantly decreased lever pressing responses induced by both cue and ketamine priming. (Figure 2A).

Subsequent to the discovery that D-serine (100 mg/kg) significantly decreased ketamine-seeking behavior induced by cue and ketamine priming, we further determined the effects of D-serine at a lower dose (30 mg/kg). A two-way repeated ANOVA unveiled the significant main effects of cue (F_1,5_ = 93.964, *p* < 0.001), D-serine treatment (F_1,5_ = 18.718, *p* < 0.01), and a significant interaction between cue and D-serine treatment (F_1,5_ = 14.199, *p* < 0.05) (Figure 2B).

In the context of ketamine priming-induced reinstatement, administering D-serine at 30 mg/kg did not lead to a reduction in lever responses (ketamine priming: F_1,5_ = 19.213, *p* < 0.01; D-serine treatment: F_1,5_ = 0.914, *p* = 0.38; and interaction between ketamine priming and D-serine treatment: F_1,5_ = 0.141, *p* = 0.723). *Post hoc* tests indicated that while D-serine (30 mg/kg) significantly reduced cue-induced lever pressing responses, it did not have the same effect on ketamine priming-induced responses.

### 2.4. Effects of Sarcosine on Reinstatement of Ketamine Seeking in Response to Cues and Drug Exposure

The effects of sarcosine at doses of 30 and 100 mg/kg on the reinstatement of ketamine-seeking behavior are depicted in Figure 2C,D. The data on cue- and ketamine-priming-induced reinstatement were analyzed using two-way repeated ANOVAs. For cue-induced reinstatement, the analysis showed significant effects of cue (F_1,10_ = 53.325, *p* < 0.001), sarcosine treatment (F_1, 10_ = 36.731, *p* < 0.001), and the interaction between cue and sarcosine treatment (F_1,10_ = 14.199, *p* < 0.001). For drug-priming-induced reinstatement, the analysis showed significant effects of drug priming (F_1,10_ = 115.22, *p* < 0.001), sarcosine treatment (F_1,10_ = 15.817, *p* < 0.001), and the interaction between ketamine priming and sarcosine treatment (F_1,10_ = 13.076, *p* < 0.01). *Post hoc* comparisons revealed that sarcosine administration (at both doses) reduced cue-induced reinstatement, whereas a higher dose of sarcosine (100 mg/kg) was required to reduce ketamine priming-induced reinstatement of the extinguished lever-press responding.

### 2.5. Effects of D-Serine and Sarcosine on Sucrose Self-Administration under a PR Schedule

A higher dose of D-serine and sarcosine (100 mg/kg) was applied to assess the response to sucrose pellets under the same PR schedule as ketamine reinforcement. There was no significant effect of D-serine and sarcosine on the breakpoints, the number of sucrose pellets, and lever responses, obtained as shown in Figure 3.

### 2.6. Effects of D-Serine and Sarcosine on Cue- and Sucrose-Induced Reinstatement of Sucrose Seeking

Figure 4 illustrates the impact of D-serine and sarcosine at a dose of 100 mg/kg on the reinstatement of sucrose-seeking behavior. The data on cue- and sucrose pellet-induced reinstatement were analyzed using two-way repeated ANOVA. For the effects of D-serine on cue-induced reinstatement, the analysis showed no significant effect of cue (F_1,5_ = 6.258, *p* = 0.054), D-serine treatment (F_1,5_ = 0.051, *p* = 0.83), and the interaction between cue and D-serine treatment (F_1,5_ = 0.0135, *p* < 0.912). Similarly, there was no significant effect of cue (F_1,5_ = 3.726, *p* = 0.11), sarcosine treatment (F_1,5_ = 2.986, *p* = 0.145), and the interaction between cue and sarcosine treatment (F_1,5_ = 0.985, *p* = 0.366). These results suggest that cue was not capable of inducing reinstatement of sucrose seeking. In addition, neither D-serine nor sarcosine had a significant effect on cue-induced reinstatement. The analysis of the effects of D-serine on sucrose priming-induced reinstatement showed a significant effect of sucrose priming (F_1,5_ = 76.904, *p* < 0.001), but no significant effect of D-serine treatment (F_1,5_ = 1.361, *p* = 0.296) or the interaction between cue and D-serine treatment (F_1,5_ = 0.36, *p* = 0.575). In the case of sarcosine treatment, there was a significant effect of sucrose priming (F_1,5_ = 29.927, *p* < 0.01), but no significant effect of sarcosine treatment (F_1,5_ = 0.519, *p* = 0.503) or the interaction between cue and sarcosine treatment (F_1,5_ = 0.958, *p* = 0.373). These results suggest that sucrose priming significantly induced the reinstatement of sucrose-seeking behavior. However, neither D-serine nor sarcosine had a significant effect on sucrose priming-induced reinstatement.

The analysis of the effects of D-serine on sucrose priming-induced reinstatement showed a significant effect of sucrose priming (F_1,5_ = 76.904, *p* < 0.001), but no significant effect of D-serine treatment (F_1,5_ = 1.361, *p* = 0.296) or the interaction between cue and D-serine treatment (F_1,5_ = 0.36, *p* = 0.575). In the case of sarcosine treatment, there was a significant effect of sucrose priming (F_1,5_ = 29.927, *p* < 0.01), but no significant effect of sarcosine treatment (F_1,5_ = 0.519, *p* = 0.503) or the interaction between cue and sarcosine treatment (F_1,5_ = 0.958, *p* = 0.373). These results suggest that sucrose priming significantly induced the reinstatement of sucrose-seeking behavior. However, neither D-serine nor sarcosine had a significant effect on sucrose priming-induced reinstatement.

### 2.7. Effects of D-Serine and Sarcosine on Locomotor Activity

To assess the effect of D-serine or sarcosine on basal locomotor activity in a novel open field, a between-subjects design was employed for comparison. A two-way mixed design ANOVA revealed that administration of D-serine (D-serine treatment: F_1,66_ = 0.024, *p* = 0.88; Time: F_11,66_ = 19.513, *p* <0.001; D-serine × Time: F_11,66_ = 1.532, *p* = 0.141) or sarcosine (sarcosine treatment: F_11,66_ = 0.839, *p* = 0.402; Time: F_11,66_ = 14.828, *p* <0.001; sarcosine × Time: F_11,66_ = 1.2, *p* = 0.409) at 100 mg/kg did not affect the locomotor activity of rats (Figure 5).

## 3. Discussion

The aim of this study was to investigate whether modulation of the NMDAR glycine binding site could reduce the abuse liability of ketamine and be a potential treatment for ketamine use disorder. Two modulators for the NMDAR glycine binding site, D-serine and sarcosine, significantly reduced the motivated behavior for ketamine under a PR schedule in the rat intravenous ketamine self-administration paradigm. Furthermore, D-serine and sarcosine effectively suppressed the reinstatement of ketamine-seeking behaviors induced by cues and ketamine priming.

An effective pharmacological treatment for addiction should be capable of reducing the reinforcing effect and reinstatement of the abused drug, without affecting the reinforcement of natural rewards. The results of this study showed that D-serine and sarcosine significantly reduced PR breakpoints, the motivational measures using the PR schedule, and the reinstatement of ketamine-seeking (relapse-like) behavior. However, they did not affect motivation for sucrose and sucrose-seeking behavior. Therefore, targeting the NMDAR glycine binding site could be a promising approach for treating ketamine use disorder.

Most of the significant clinical effects of ketamine, including anesthesia, analgesia, and antidepressant properties, have been primarily attributed to its interaction with the NMDAR. It has been suggested that the rewarding actions of NMDAR antagonists such as PCP and MK-801, are not dependent on dopamine and are associated with the inhibition of medium spiny neurons in the nucleus accumbens [11]. An alternative explanation for the rewarding actions of ketamine has been proposed, which involves the inhibition of NMDARs in GABA neurons located in the ventral tegmental area (VTA). This inhibition enhances the activity of dopamine neurons in the VTA, resulting in a brief burst of dopamine release in the nucleus accumbens and prefrontal cortex [24,25]. Moreover, elevated phosphorylation of the GluN2B subunit of the NMDAR has been associated with ketamine self-administration, suggesting the involvement of NMDAR in ketamine’s addictive properties [26].

This study examined and supported the notion that modulation of the NMDAR glycine binding site can reverse the reinforcing efficacy of ketamine. Our results demonstrated that D-serine and sarcosine administered at a dose of 100 mg/kg decreased the breakpoints under a PR schedule, indicating reduced motivation for ketamine. Both D-serine and sarcosine presumably act on the NMDAR glycine binding site along with glutamate to modulate NMDAR activity. They also regulate the trafficking of NMDAR. Additionally, when the NMDAR co-agonist site is occupied by glycine or D-serine at high concentrations, it primes the NMDAR for endocytosis [23]. These glutamate-independent modulatory effects of co-agonists are specific to certain types of neurons [27] and NMDAR subtypes [22], which may be associated with their ability to reduce the abuse liability of ketamine.

The repeated use of ketamine for the treatment of depression has raised significant concerns regarding its potential for misuse, leading to the development of addiction. Therefore, there is a need to develop pharmacological agents that can enhance the antidepressant effects of ketamine while minimizing its abuse potential. Interestingly, both D-serine and sarcosine have also demonstrated antidepressant effects in pre-clinical models [28,29] and in human trials [30]. Although it remains unclear whether D-serine and sarcosine could enhance the antidepressant effect of ketamine, their ability to counteract the abuse liability of ketamine suggests that they may be suitable for combined treatment with ketamine in depressed patients.

Relapse often occurs when individuals encounter stimuli or cues previously associated with drug use. Our results demonstrated that cue and ketamine priming injection could reinstate the extinguished lever response for ketamine. D-serine and sarcosine at 30 and 100 mg/kg reduced the cue-induced reinstatement of ketamine seeking. A higher dose of D-serine and sarcosine (100 mg/kg) was required to suppress the ketamine priming-induced reinstatement. These results suggest that D-serine and sarcosine are more effective for cue than drug priming-induced reinstatement. The brain circuits recruited for the reinstatement of ketamine remain to be determined, which should be different from sucrose and other drugs of abuse. D-serine and sarcosine did not affect the sucrose priming-induced reinstatement of seeking. Sarcosine treatment has been shown to suppress the expression and facilitate the extinction of cocaine-associated memory in cocaine-induced conditioned place preference [31]. It is of interest to investigate whether D-serine and sarcosine could also suppress the reinstatement of cocaine and other drugs of abuse.

Biological sex strongly modulates drug self-administration behavior in rodents [32]. The effects of sex and the estrus cycle on intermittent ketamine self-administration have been investigated [33]. The results indicated that female rats trained during the diestrus phase were unable to maintain ketamine self-administration and did not exhibit reinstatement to ketamine-associated cues. However, male rats and female rats trained during the proestrus phase showed reinstatement to ketamine-associated cues. These findings underscore the importance of considering sex differences and hormonal factors when studying the response to NMDAR modulators aimed at reducing the reinforcing effects and reinstatement of ketamine-seeking behavior. It is worth noting that the present study only utilized male rats, and thus, it is critical to incorporate sex-specific approaches in future research. By doing so, we can gain a more comprehensive understanding of how sex influences the response to NMDAR modulators and develop targeted interventions that account for these sex differences.

D-serine is an NMDAR activator at the glycine site but actually inhibits non-canonical NMDAR [34]. D-serine has been demonstrated to be effective in treating schizophrenia [35], mood disorders [36], and cognitive enhancement in healthy subjects [37] which is mediated by activation of canonical NMDAR. In addition, D-serine reduced compulsion-like alcohol intake and this effect has been attributed to its inhibition of non-canonical NMDARs [38]. On the other hand, sarcosine does not affect non-canonical NMDAR and has similar effects to D-serine on schizophrenia [39]. Both D-serine and sarcosine reduced the breakpoints and reinstatement for ketamine. It is likely that canonical NMDAR plays a more important role than non-canonical NMDAR in their effects on ketamine abuse liability and relapse-like drug-seeking behavior.

Studies have shown that NMDAR modulators targeting the glycine binding site can improve schizophrenia-like behaviors and biochemical changes induced by ketamine [40,41]. Our findings further support the effectiveness of D-serine and sarcosine in reducing motivation and relapse-like drug-seeking behavior associated with ketamine use. D-serine and sarcosine are readily available through nutritional supplement vendors, are well-tolerated, and have minimal adverse effects. These results underscore the potential of NMDAR modulators that target the glycine binding site as an add-on supplement for managing treatment-resistant depression and reducing the abuse liability of ketamine. Furthermore, NMDAR modulators may also prove beneficial in reducing cravings and relapse in patients suffering from ketamine use disorder.

## 4. Materials and Methods

### 4.1. Animals

Seventy-two male Sprague–Dawley rats weighing between 225 and 250 g were obtained from LASCO Charles River Technology (Taiwan) and housed in an animal care facility with a 12-h light/dark cycle. The rats were given unrestricted access to water. Except after surgery, food intake was restricted during the studies. Following surgery, rats were fed ad libitum for 7 days, and 15 g of lab chaw was administered immediately after each daily drug self-administration session.

The Institutional Animal Care and Use Committee of the National Health Research Institutes reviewed and approved all animal care procedures (NHRI-IACUC-108044-A). The number of animals assigned to each experiment was shown in Table 1.

### 4.2. Drugs

Ketamine HCl, D-serine, and sarcosine (Sigma, St. Louis, MO, USA) were dissolved in saline. D-serine and sarcosine were administered via intraperitoneal injection.

### 4.3. Ketamine Self-Administration Procedures

The self-administration procedures were conducted in sound-attenuating cubicles equipped with operant chambers (32 × 25 × 34 cm, Med Associates, Inc., Latham, NY, USA) with a ventilation fan and connected to a computerized data collection program. Inside each chamber, two retractable levers were installed with a yellow stimulus light above each lever.

The ketamine self-administration procedures were identical to those previously used in our previous study [42]. Briefly, prior to ketamine self-administration, sucrose pre-training proceeded. After food restriction (5 g/day) for 48 h, rats underwent training to press the lever for sucrose (45 mg pellet; Product# F0023, Bio-Serv, Flemington, NJ, USA), under a fixed ratio (FR) 1 until they could earn 100 sucrose pellets during 1 h for three consecutive days.

Following sucrose pre-training, rats underwent intravenous catheterization surgery. A silastic tubing was implanted into the external jugular vein, with the other end connected to an injection port in a harness. A daily flush was conducted with a solution comprising of antibiotic Baytril (2.5%; Bayer) and heparinized saline (50 IU/mL).

During ketamine self-administration training, rats were given access to daily 2 h sessions, where pressing the active lever triggered the syringe pump to administer ketamine (0.5 mg/kg/infusion, 4 s) under an FR1 TO20, followed by an FR2 TO20 schedule until the response patterns were stable. Concurrent with each ketamine infusion, the stimulus light was illuminated for 20 s. The self-administration training sessions continued until the response patterns reached a stable and consistent state.

### 4.4. A Progressive Ratio (PR) Schedule of Reinforcement

After the completion of FR2 training, a progressive ratio (PR) schedule was implemented. The number of lever presses required to obtain a single infusion was determined by: 5 × e^(infusion number × 0.2)^ − 5 (i.e., 1, 2, 4, 6, 9, 12, 15, 20, 25, 32, etc.) [43]. If animals fail to obtain another infusion within 1 h, the PR schedule was terminated automatically. The final ratio of the last infusions obtained was considered the breakpoint. The lever responses, infusion number, and breakpoints were recorded.

In order to evaluate the effects of D-serine and sarcosine (30 or 100 mg/kg, i.p.) on the breakpoint, a within-subjects design was employed. D-serine or sarcosine was administered 2 h or 30 min prior to the initiation of self-administration sessions, respectively. Administration of D-serine led to a slow rise in both extracellular and intracellular concentrations within the brain due to its weak diffusion through the blood-brain barrier [44]. Therefore, D-serine was administered 2 h prior to the assessment. To minimize the residual effect of testing compounds, there was a gap of 2–3 days between test days. On the days between the test days, the same PR schedule was conducted to maintain the animals’ familiarity and consistent response patterns.

### 4.5. Cue- and Drug Priming-Induced Reinstatement of Ketamine Seeking

After achieving stability in responding to the FR2 schedule, animals were subjected to daily 2-h extinction sessions, where pressing the active lever no longer resulted in any programmed reinforcement consequences such as the presentation of cue light or activation of the syringe pump. The extinction criteria were considered met when the rats performed with the number of active levers presses below 20% of the final FR2 sessions.

Following the extinction phase, a within-subject design was employed to test the effects of D-serine and sarcosine on cue- and ketamine-induced reinstatement (each lasting 2 h). A re-extinction session was conducted between reinstatement tests to restore the extinguished lever response. In the cue-induced reinstatement test, the presentation of cue light was accompanied by successful active lever-press responses and throughout the timeout period, without any ketamine infusions.

For the ketamine-primed reinstatement test, the rats received the priming dose of ketamine (10 mg/kg, i.p.) immediately prior to the start of the 2-h session. During the session, active lever responses that did not result in cue light presentations or drug infusions were recorded.

### 4.6. Sucrose Self-Administration Procedures

The effects of D-serine and sarcosine on breakpoints for sucrose and sucrose-seeking behavior were tested in rats that did not undergo catheter implantation. After sucrose pre-training, rats were allowed to access 2 h daily sessions. They engaged in sessions under an FR1 TO20 s (5 days), followed by an FR2 TO20 s (5 days), where pressing the active lever to deliver a single sucrose pellet. Subsequently, animals were subjected to a PR schedule.

Another set of well-trained rats underwent the daily 2 h extinction session until they reached the extinction criteria (<20% responses of last FR2 session). The effects of D-serine and sarcosine both administered at a dose of 100 mg/kg on cue and sucrose priming-induced reinstatement of sucrose seeking were evaluated.

### 4.7. Open Filed Test

We determined whether the effective dose (100 mg/kg) of D-serine and sarcosine could affect spontaneous locomotor activity in the open field. Prior to the administration of D-serine, sarcosine, or the vehicle, the rats were introduced into an open field chamber (42 × 42 × 30 mm, Animal Activity Monitoring System, AccuScan Instruments, Inc., Trabue Rd Columbus, OH, USA) for 60 min to habituate. The distances traveled were monitored for 120 min after injection of D-serine, sarcosine, or the vehicle.

### 4.8. Data Analysis

The effects of D-serine and sarcosine on ketamine and sucrose self-administration under a PR schedule, as well as the reinstatement of ketamine seeking were assessed by the within-subjects design. The number of lever responses, breakpoints, and the number of ketamine infusions under the PR schedule were analyzed using paired *t*-tests or repeated one-way ANOVAs. In the reinstatement experiments, a two-way repeated measures ANOVA was employed. Newman–Keuls tests were used for *post-hoc* comparisons. The distances traveled during the locomotor activity test were analyzed using a two-way mixed design ANOVA, with time being the within-subject factor. All data are expressed as the mean ± SEM.

## Figures and Tables

**Figure 1 pharmaceuticals-16-00812-f001:**
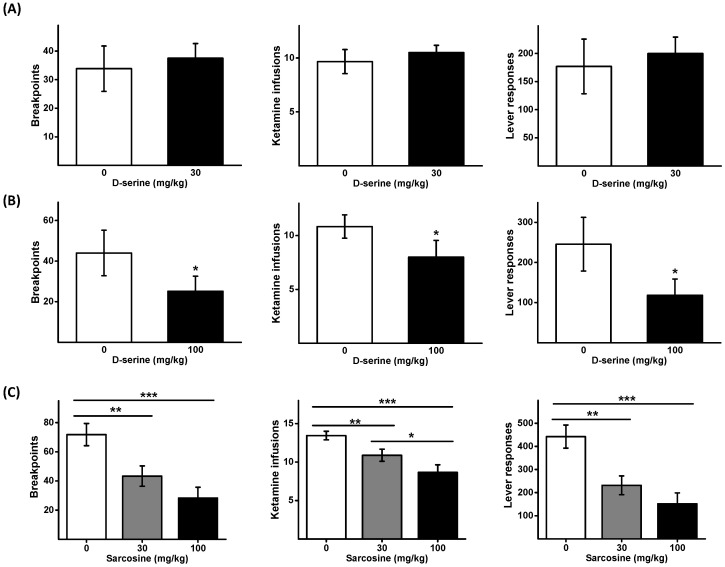
Effects of D-serine and sarcosine on the motivation to self-administer ketamine. Animals received either D-serine (30 mg/kg) or vehicle (**A**), and D-serine (100 mg/kg) or vehicle (**B**) 120 min before the initiation of ketamine self-administration under a PR schedule. Another set of animals received sarcosine (30 or 100 mg/kg) or vehicle (**C**) 30 min before the self-administration commenced. The numbers of breakpoints, lever responses, and ketamine infusions are presented as mean ± SEM (*n* = 6). * *p* < 0.05, ** *p* < 0.01, *** *p* < 0.001 compared with vehicle controls.

**Figure 2 pharmaceuticals-16-00812-f002:**
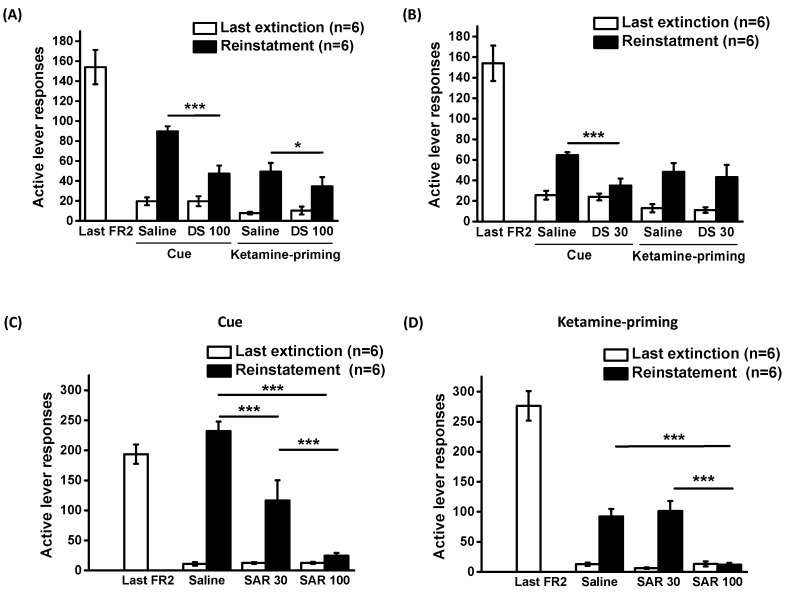
Effects of D-serine and sarcosine on reinstatement of ketamine seeking. Following extinction training, D-serine (100 mg/kg) or vehicle (**A**) and D-serine (30 mg/kg) or vehicle (**B**) were administered to the rats 120 min before the reinstatement tests. Sarcosine (0, 30 or 100 mg/kg) was given 30 min before reinstatement induced by cue (**C**) and ketamine priming (**D**). The numbers of active lever responses during the last session training under a FR2 schedule, the final session of each extinction procedure and reinstatement induced by cue or ketamine priming were illustrated. The data are displayed as mean ± SEM (*n* = 6). * *p* < 0.05, *** *p* < 0.001 compared with vehicle controls. DS: D-serine; SAR: sarcosine.

**Figure 3 pharmaceuticals-16-00812-f003:**
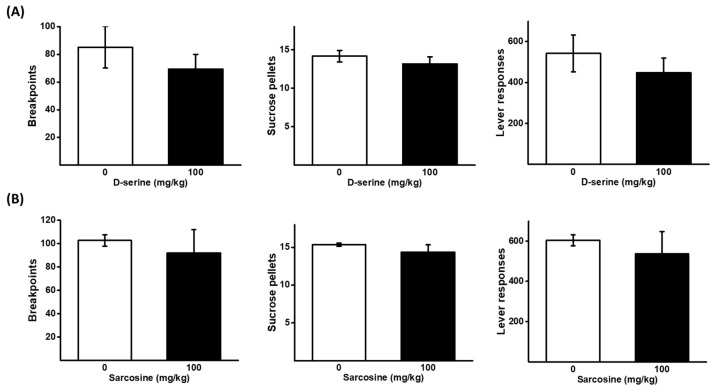
Effects of D-serine and sarcosine on the motivation for sucrose. D-serine at a dose of 100 mg/kg or vehicle (**A**), and sarcosine at a dose of 100 mg/kg or vehicle (**B**), were administered either 120 min or 30 min prior to the self-administration of sucrose pellets under a PR schedule, respectively. The numbers of breakpoints, sucrose pellets received, and lever responses, were recorded. The data are presented as mean ± SEM (*n* = 6).

**Figure 4 pharmaceuticals-16-00812-f004:**
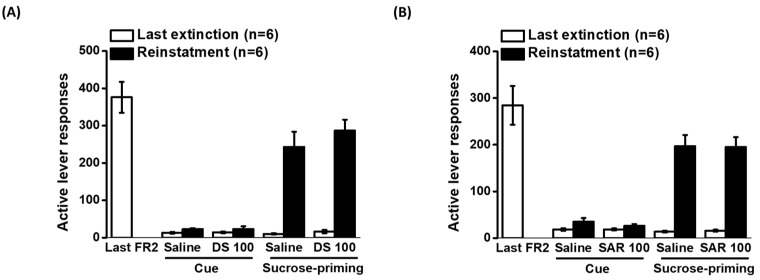
Effects of D-serine and sarcosine on cue- and sucrose-induced reinstatement of sucrose seeking. Rats were trained self-administration for food reward with sucrose pellets. Following extinction procedures, D-serine (100 mg/kg) or vehicle (**A**) and sarcosine (100 mg/kg) or vehicle (**B**) were administered 120 min and 30 min before cue- and sucrose priming-induced reinstatement, respectively. The numbers of active lever responses during the last session of training, the final extinction session, and reinstatement induced by cue or sucrose were displayed. The data are presented as mean ± SEM (*n* = 6). DS: D-serine; SAR: sarcosine.

**Figure 5 pharmaceuticals-16-00812-f005:**
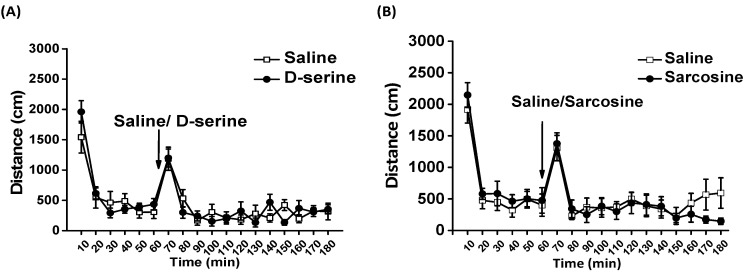
Effects of D-serine and sarcosine on open field locomotor activity. Basal locomotor activity was monitored for 60 min, after which D-serine (100 mg/kg) or saline (**A**) and sarcosine (100 mg/kg) or saline (**B**) were administered (as indicated by the arrow), and locomotor activity was measured for another 120 min. The distance traveled of rats for each 10-min interval is displayed as mean ± SEM (*n* = 6).

**Table 1 pharmaceuticals-16-00812-t001:** The number of animals assigned to each experimental group.

Test	Self-Administration								Locomotor Activity	
	Ketamine				Food					
Procedure	PR		Reinstatement		PR		Reinstatement			
	D-serine	Sarcosine	D-serine	Sarcosine	D-serine	Sarcosine	D-serine	Sarcosine	D-serine	Sarcosine
Numbers	6	6	6	6	6	6	6	6	12	12

## Data Availability

Data is contained within the article.

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
