# Peer review of "NMDA Receptor Glycine Binding Site Modulators for Prevention and Treatment of Ketamine Use Disorder"

_pharmaceuticals, 2023, doi:10.3390/ph16060812_

Round 1

Reviewer 1 Report

1.   The authors did not provide the route of administration of D-serine and sarcosine. Because different administration methods will affect the interpretation of the results, the author should state the route of administration of D-serine and sarcosine.

2.   In this study, the author writes in lines 348-352: “D-serine or sarcosine was administered 2 hr or 30 min prior to the initiation of self-administration sessions, respectively. Administration of D-serine resulted in a slow increase in both extracellular and intracellular concentrations in the brain due to its weak diffusion through the blood-brain barrier [40]. Therefore, D-serine was administered 2 h prior to the assessment.”  But in fact, Pernot’s research (reference 40) indicates that an intraperitoneal D-serine administration increases brain extra- and intra-cellular concentrations despite weak diffusion through the blood-brain barrier. In particular, D-serine rose rapidly in the CSF after peripheral (intraperitoneal) administration. The authors should again carefully explain the reason for the difference in the timing of administration of the two drugs.

3.   I recommend that the authors detect the concentration of D-serine or sarcosine in the brain after drug administration. That would better explain the effects of the drugs.

4.  Figure 4 lacks statistically significant symbols.

5.   Please delete “its” in line 223.

Reviewer 2 Report

In this paper, the authors show how two NMDAR modulators, sarcosine and D-serine, could mitigate the ketamine seeking behaviour in male Sprague-Dawley rats. This is an important topic and the results obtained by the authors are potentially interesting. I have a few comments which could improve the manuscript, as follows:

Main points: 

-          Figure 5, and point 2.7:  In the related text, the authors should tone down their conclusion that “sarcosine…did not affect the locomotor activity” by replacing with “basal locomotor activity”, as correctly indicated in the legend. The term “locomotor activity” has a very broad spectrum of interpretations and the test performed by te authors does not cover them all.

How many animals in total have been used in this study? How many rats were estimated to be necessary to carry out this study protocol to begin with? Each figure reports n = 6, but it is not clear if the same rats underwent different procedures or whether the groups were all separated. Sometimes, it is generally stated “a separate group of rats….” but this is still quite confusing. A table, showing the total number of animals and the different experimental groups that were created would greatly simplify the interpretation of this aspect.

 Discussion: The authors have chosen to use only male rats. This has been a consolidated habit for years but the study of sex differences in rodent models of addictive behavior (Radke et al., 2021) is also a matter of great importance. Please add a paragraph in the discussion to inform the readers about this possible limitation of the study.

Further, the authors state that “This study is the first to examine and support the notion that modulation of the NMDAR glycine binding site can reverse the reinforcing efficacy of ketamine”. Please tone down also this statement, since similar results have already been published by other research groups. (e.g. Kumar et al., 2023; Peyrovian 2019). The authors face the issue from a different perspective, but nonetheless previous similar literature should be mentioned while discussing their new findings.

Minor points:

-       "Result" Heading: please change to “results” (line 90)

 Figures: the resolution of the figures embedded in the text is very low. If they are to be considered definitive, please try to increase the quality of the graphics.
